# Patients' and healthcare professionals' perspectives towards technology-assisted diabetes self-management education. A qualitative systematic review

Sneha Rajiv Jain[1], Yuan Sui[1], Cheng Han Ng[1], Zhi Xiong Chen[2,3], Lay Hoon Goh[4], Shefaly Shorey[5]*

1 Yong Loo Lin School of Medicine, National University of Singapore, Singapore, Singapore, 2 Department of Physiology, Yong Loo Lin School of Medicine, National University of Singapore, Singapore, Singapore, 3 Centre for Medical Education, Yong Loo Lin School of Medicine, National University of Singapore, Singapore, Singapore, 4 Division of Family Medicine, Yong Loo Lin School of Medicine, National University of Singapore, Singapore, Singapore, 5 Alice Lee Centre for Nursing Studies, Yong Loo Lin School of Medicine, National University of Singapore, Singapore, Singapore

* nurssh@nus.edu.sg

## Abstract

### Introduction

Diabetes self-management education is a key aspect in the long-term management of type 2 diabetes. The patient and healthcare professional (HCP) perspective on the use of technology-assisted DSME has yet to be studied. Hence, the objective of this study was to better understand the factors that facilitate or hinder the adoptions of such education by adults with type 2 diabetes and their HCPs.

### Methods

We systematically searched five databases (Medline, Embase, CINAHL, Web of Science Core Collection, and PsycINFO) until August 2019. The search included qualitative and mixed-method studies that reported the views of patients and HCPs regarding features, uses, and implementations of technology-assisted DSME. Data were synthesized through an inductive thematic analysis.

### Results

A total of 13 articles were included, involving 242 patients, ranging from 18 to 81 years and included web-based, mobile application, digital versatile disc (DVD), virtual reality or tele-health interventions. Patients and HCPs had mixed views towards features of the technology-assisted interventions, with patients' personal qualities and HCPs' concerns affecting uses of the interventions. Patients generally preferred technologies that were easy to access, use, and apply and that had reliable information. Patients' ambitions motivated them, and personal attributes such as poor competence with technology, poor literacy, and language barriers acted as barriers. Patients especially liked the peer support that they

**Data Availability Statement:** All relevant data are within the paper and its Supporting Information files.

**Funding:** This paper is funded independently by one of the authors, Goh Lay Hoon.

**Competing interests:** The authors have declared that no competing interests exist.

**Abbreviations:** HbA1c, Haemoglobin A1c; DSME, Diabetes Self-Management Education; HCP, Healthcare Professional; PRISMA, Preferred Reporting Items for Systematic Reviews and Meta-Analyses; sENTREQ, Enhancing transparency in reporting the synthesis of qualitative research.

received but did not like it when there was no regulation of advice on these platforms. HCPs believed that while the interventions were useful to patients, they faced difficulties with integration into their clinical workflows.

## Conclusion

This review explored the features of technology-assisted diabetes self-management education interventions that enhanced positive patient engagements and the negative aspects of both the platforms and the target groups. Technical support and training will be effective in managing these concerns and ensuring meaningful use of these platforms.

## Introduction

Diabetes is a complex disease. Its successful management is as much of an art as it is of a science. While the science behind diabetes is a body of well-understood and stable knowledge, the art of managing diabetes remains a dynamic process that requires constant understanding and updates of the interplay between psychological, social, economic, cultural and behavioural factors affecting patients, healthcare professionals (HCPs), and the society [1, 2]. This is further complicated by the need to blend with emerging science on diabetes and technological breakthroughs in delivering patient education.

As a chronic disease, the responsibility for successfully managing diabetes cannot lie on HCPs alone as it requires the co-sharing of responsibility between patients and various HCPs [3, 4]. Inherently, type 1 and type 2 diabetes are very different. While type 1 diabetes is largely of childhood or juvenile-onset with a genetic component and a lower prevalence [5, 6], type 2 diabetes has a larger prevalence in adult and elderly patients as well as a link to insulin resistance [7, 8]. In terms of management, type 1 diabetes is mainly treated pharmacologically with exogenous insulin [6], while type 2 diabetes consists of education, lifestyle modifications, and oral hypoglycemic agents, all of which require strict adherence to ensure their effectiveness [9]. In this regard, diabetes self-management education (DSME) becomes a very important component of diabetes care since it provides a foundation to help people navigate their decisions and activities in view of their chronic conditions [10]. This is especially the case for type 2 diabetics since it requires one to make complex self-care decisions daily [10]. DSME involves the continuous transfer and facilitation of skills and knowledge for empowering patients with the abilities to self-care throughout their lifetimes, starting from their diagnoses [11, 12]. This can include information and facts about the disease, how to self-monitor blood glucose and its importance, how to prevent or identify and manage unstable glucose levels and other complications, and how to access information updates and reminders on screenings for diabetes-associated complications [13].

The American Association of Diabetes Educators 7 (AADE7) Self-Care Behaviors™, a framework that helps patients to adopt healthy habits, be compliant with medication, and cope with diabetes-related issues, is widely incorporated in most DSME efforts [13, 14]. Indeed, DSME has been shown to enhance patient outcomes in terms of reducing mortality and complications, and improving quality of life through lowering of glycated hemoglobin (HbA1c) levels, having better control of blood pressure and weight management, and successful implementation of lifestyle changes [13]. This is due to patients having more knowledge, hence being able to take better control of their diabetes by making more informed decisions [15].

This has allowed HCPs in the collaborative care model to more effectively manage their patients [16].

As we head into the fourth industrial revolution, the explosion of disruptive technologies into the scene is transforming the way education is being delivered faster than ever before, and DSME is no exception [17]. Such innovative and novel disruptive technologies have transfomed the traditional face-to-face counselling and delivery of self-management information to patients [18]. They do so by using information and communications technology to create an entirely new avenue for HCPs and patients to acquire information to facilitate care-coordination, promote health literacy and patient activation, and increase accessibility [18]. This ranges from delivering DSME through simple websites to more sophisticated cloud-based platforms [17]. As with many technological advances, availability often may not translate to adoptions of the technology by patients or HCPs for various reasons. On the other hand, technology-assisted efforts may or may not lead to better patient experiences or improved clinical outcomes, relative to non-technology-based interventions [19].

There is a growing diversity of technology-assisted DSME platforms, such as mobile health applications, text messaging systems, gaming systems, internet-based interventions, web-based learning platforms, and computer-assisted education programmes [20]. Previous literature has shown that increased uses of technological interventions, especially web-based interventions, were associated with greater improvements in outcomes, such as significantly decreased HbA1c, decreased postprandial glucose levels, and improved diabetes control [21, 22]. However, there has not been any qualitative systematic review that examined patients' and HCPs' perspectives of technology-assisted DSME for type 2 diabetics. While current reviews have examined the effectiveness of such interventions, it is unclear whether there are any barriers or facilitators to their adoptions, how patients may interact with the technologies, how relevant are the contents, how other stakeholders such as nurses and other HCPs in the caregiving network are connected, and interactions between patients' 'soft' needs (e.g. motivations, beliefs, and interests) and technology. By using a qualitative, person-based approach for such technological interventions, it is possible to examine user experiences and to find ideals for users to follow to create opportunities for intended behavior changes [23]. By systematically synthesizing the perspectives of stakeholders, it is possible to use this feedback to create new or improve existing interventions, supplementing the theory behind an intervention's development [24].

Hence, this qualitative systematic review seeks to answer these questions through existing literature on patients' and HCPs' perceptions towards technology-assisted DSME. The aim is to derive insights that will help in the development of more effective and personalized technology-assisted DSME platforms that can be contextualized to any healthcare setting.

## Materials and methods

The Preferred Reporting Items for Systematic Review and Meta-analysis (PRISMA) [25], and the Enhancing Transparency in Reporting the Synthesis of Qualitative Research (ENTREQ) statements were abided in the synthesis of this review [26].

### Search strategy and screening

The following electronic databases were searched from inception to 28 August 2019: Medline, Embase, CINAHL, Web of Science Core Collection, and PsycINFO. The search algorithm is attached in S1 Appendix. Review databases such as PROSPERO, Cochrane, *Epistemonikos* and *McMasters Health evidence* were checked on 9 April 2020 to ensure that such a review had not been conducted previously. The citations were downloaded, and duplicates were removed with the EndNote X9 software. Two authors (SRJ and SY) independently screened the citations

manually using EndNote, and those that did not fulfil the inclusion criteria were excluded, after which a full text review was conducted. Those that met the inclusion criteria were included in this article, with differences being solved by consensus. The PRISMA flowchart in Fig 1 depicts the flow process of the review.

Studies were included if they 1) involved technology-assisted DSME, 2) involved only adults with type 2 diabetes mellitus or their HCPs, 3) examined views of uses of their interventions, and 4) were of mixed methods or qualitative (focus group, narrative, in-depth interview). Adults were defined as anyone over the age of 18 years, and searches were limited to the

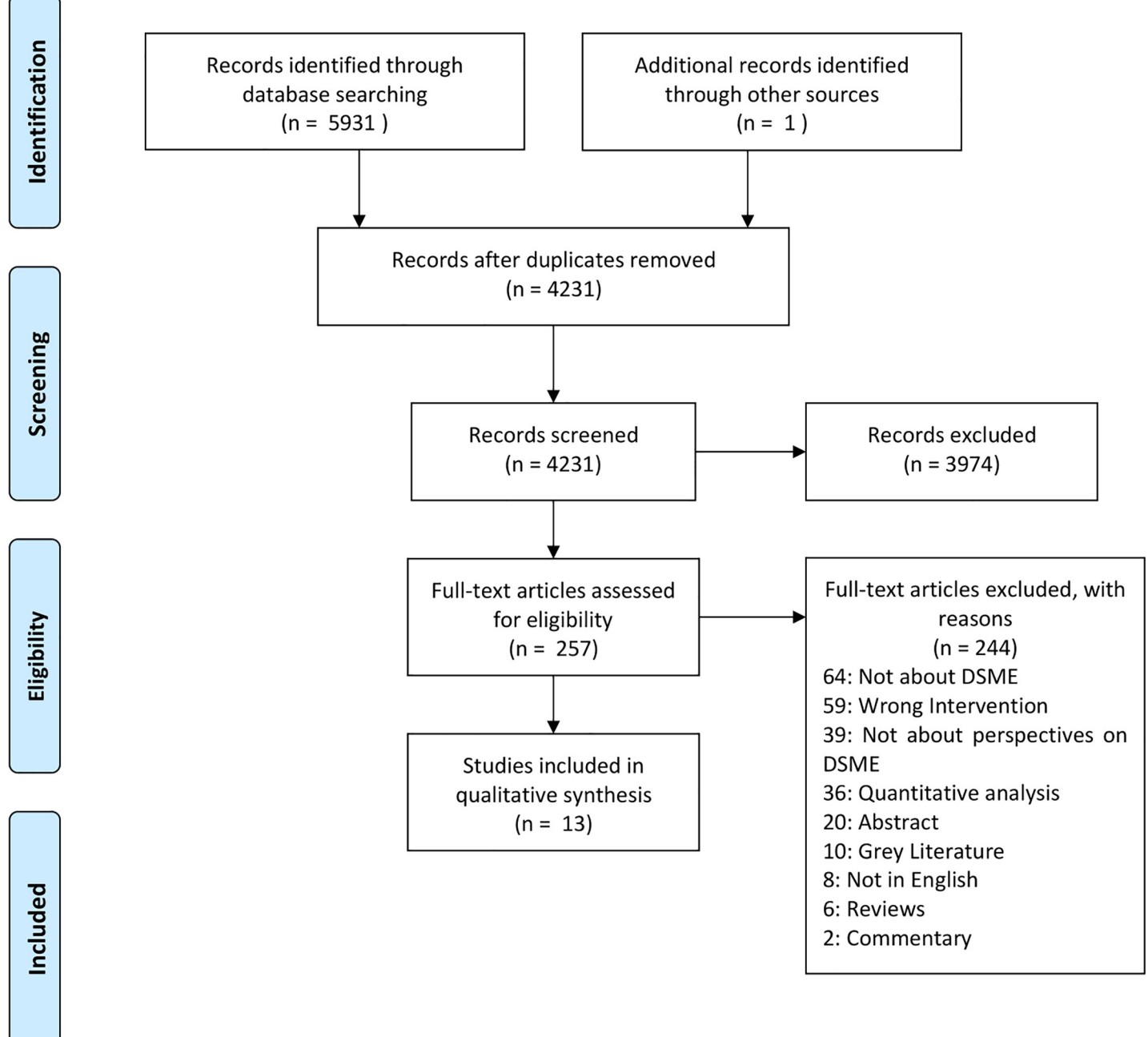

**Fig 1. PRISMA flowchart of search results.**

English language. Only original, peer-reviewed papers were considered. Commentaries, reviews, conference abstracts, dissertations and thesis papers were excluded. The screening of the titles and abstracts was carried out by the authors (SRJ, SY, CHN, CZX and GLH). Two authors (SRJ and SY) independently conducted a full-text review using Excel, and discrepancies on the inclusions were discussed and reached a until a consensus was reached, with input from senior authors within the team (CHN and GLH).

## Data extraction and synthesis

Two authors (SRJ and SY) independently read the selected articles and recorded and extracted data using a structured proforma on Excel, after which codes were compared to ensure comparability. The structured proforma included the origin and year of publication, methodology, demographics (sample size, gender, and age) of the participants, and primary findings in the Results section. Data were thematically synthesized using Thomas and Harden's methodology [27] using three stages, namely, line-by-line coding, the derivation of descriptive themes, and the derivation of analytical themes. Articles were individually coded by the two authors (SRJ and SY), and the primary codes were discussed, compared, and analyzed before synthesizing the final primary codes. Descriptive themes were formed by grouping, reviewing, and analyzing similar overarching concepts in the primary codes. Analytical themes were derived using a thematic analysis to interpret new hypotheses and explanations beyond that of the primary research. An example of this thematic synthesis is available in S3 Appendix. Discussions between the authors (SRJ, SY, CHN, GLH, and CZX) were held for clarifications and interpretations of the primary findings and for the final synthesis' findings [27, 28].

## Quality appraisal of the included studies

A quality appraisal of the studies was conducted at the study level using the Critical Appraisal Skills Programme (CASP) Qualitative Review Checklist in order to improve the rigor of the synthesis by assessing the strength of the articles based on the validity of the recruitment, data collection, data analysis, and results based on a clearly defined criterion [29]. The checklist was used as a guide to evaluate the validity, results, and clinical relevance of the included studies [29] by assessing the credibility, transferability, dependability, and confirmability of the qualitative research [30]. The CASP tool is recommend by the Cochrane Methods Qualitative and Implementation group [30]. Two authors (SRJ and SY) independently conducted the CASP assessment, with disagreements resolved with a third author (CHN).

## Results

The electronic search results identified a total of 4,233 abstracts and 259 articles were subjected to a full text review, of which 15 papers met the inclusion criteria [31–43]. Cumulatively, the articles included a total of 270 adult patients with type 2 diabetes mellitus and HCPs, with five studies originating from the United States, five from the United Kingdom, three from Canada and one from Iran and Belgium respectively. The patients' ages ranged from 18 to 81 years. There were five types of interventions used in the papers. Nine papers used a web-based interventions [32, 33, 35, 36, 39–44], while the others used either a mobile phone applications (apps) [38, 45], a digital versatile disc (DVD) [34], virtual reality [31], or telehealth [37]. Three studies also explored the HCPs' views on this subject [34, 37, 41]. Nine studies used semi-structured interviews [33–38, 40, 44, 45], four used focus groups [39, 41–43], while the remaining two used both methods [31, 32]. In this review, HCPs refers to practice nurses, nurse practitioners, patient care technicians, and telehealth nurses. The characteristics of the included papers are presented in Table 1.

**Table 1.  Main characteristics of the included papers.**

| Author, year | Country | Participants (number; gender (male %); age (range/mean)) | Methodology | Perspective | Intervention | Objective of the study | Details of the technology |
|---|---|---|---|---|---|---|---|
| D. K. King et al., 2012 [42] | USA | n = 30; 46.7; 58.9 | Qualitative; focus groups | Patients | Asynchronous (web-based) | To find what patients with type 2 diabetes want from electronic resources that are designed to support their diabetes self-management. | The technology-assisted DSME intervention, MyPath, tested a minimal human contact, 12-month web-based self-management intervention that was designed to provide electronic support for adults with type 2 diabetes to improve their eating, physical activity, and medication-taking behaviors. |
| S. E. Mitchell et al., 2014 [31] | USA | n = 16; 0; 40+ | Qualitative; focus groups | Patients | Asynchronous (virtual reality) | To characterize participants' experiences of a diabetes self-management education program delivered via a virtual world versus a face-to-face format. | A virtual world is a 3D, computer-based simulated environment that presents perceptual stimuli to the user who can in turn manipulate elements of the modeled world. Second Life is an example of a free, open-access, avatar-based virtual world that supports a high level of social networking and immersive interactions with information. |
| C. H. Yu et al., 2014a [32] | Canada | n = 23; 29; 40–79 | Mixed methods; focus groups, semi-structured interviews | Patients | Asynchronous (website) | To design and test a web-based self-management tool for patients with type 2 diabetes for its usability and feasibility. | This is a website focused on facilitating the management of diabetes, including optimizing vascular risk factors. Feedback, goalsetting, peer story-telling, and monitoring tools were incorporated. In order to complement patient health information-seeking behaviors, automated emails with selected content (such as tailored reminders, or new content) were sent, search algorithms to enable self-directed information retrieval were optimized, and tools to facilitate communication with HCPs were included. |
| C. H. Yu et al., 2014b [33] | Canada | n = 21; 43; 20–79 | Mixed methods; individual semi-structured interviews | Patients | Asynchronous (website) | To determine the effect of a web-based patient self-management intervention on psychological (self-efficacy, quality of life and self-care) and clinical (blood pressure, cholesterol, glycaemic control and weight) outcomes. | The Diabetes Online Companion is a self-contained diabetes self-management website that was systematically developed according to the self-efficacy theory. The website had four main components: 1) general information (static), 2) tailored information (interactive), 3) self-monitoring logs (interactive), and 4) a blog (interactive). |

(*Continued*)

**Table 1.** (Continued)

| Author, year | Country | Participants (number; gender (male %); age (range/mean)) | Methodology | Perspective | Intervention | Objective of the study | Details of the technology |
|---|---|---|---|---|---|---|---|
| N. Patel et al., 2015 [34] | UK (England) | n = 3; NA; NA | Mixed methods; face-to-face interviews, telephone interviews | Patient and HCPs (practice nurse) | Asynchronous (DVD) | To develop and pilot-test the feasibility and effectiveness of an interactive DVD about misconceptions within South Asian communities regarding insulin treatments in type 2 diabetes, for educating patients and community members and training healthcare providers. | A DVD was created with researchers who collaborated with a multidisciplinary group of staff from the diabetes education and self-management for ongoing and newly diagnosed team, including nurses and a dietician. The script was organized to acknowledge and then correct a misconception, followed by a question to test understanding. It also included a quiz at the end. |
| M. Hofmann et al., 2016 [35] | UK (England) | n = 19; 68; 41–83 | Mixed methods; cohort study; semi-structured interviews | Patients | Asynchronous (website) | To explore the impact of using a newly developed internet-based self-management intervention called Healthy Living for People with type 2 Diabetes (HeLP-Diabetes) on the psychological well-being of adults with type 2 diabetes. | HeLP-Diabetes is an internet-based self-management intervention. It takes a holistic view of self-management and addresses a wide range of patient needs, including education, lifestyle changes, medicine management, emotional management, social support with forums, and personal stories, and also addresses how patients interact and work with health professionals. Patients were each given a printed guide and had options of receiving weekly phone calls, texts, or emails to remind them to use the website. |
| J. Jafari et al., 2016 [36] | Iran | n = 9; 56; 43.3 | Qualitative, prospective; semi-structured interviews | Patients | Asynchronous (website) | To explore the educational needs and design aspects of personalized internet-enabled education for patients with diabetes in Iran. | NA |
| D. D. Maglalang et al., 2017 [38] | USA | n = 45; 38; 57.6 | Qualitative; semi-structured interviews | Patients | Synchronous (telehealth) | To assess the acceptability and cultural relevance of the PilAm Go4Health program, a culturally adapted mobile health weight-loss lifestyle intervention including virtual social networking for Filipino Americans with type 2 diabetes. | The participants initially received the PilAm Go4Health three-month intervention and were asked to 1) each wear a Fitbit accelerometer daily, 2) self-report food/calorie intakes and weights using the Fitbit diary application, and 3) participate in the private Facebook group. The research staff posted weekly healthy lifestyle education on the private Facebook site and facilitated ad hoc virtual group discussions. After three months, the participants transitioned to a three-month maintenance to continue healthy behaviors on their own. |

(*Continued*)

Table 1. (Continued)

| Author, year | Country | Participants (number; gender (male %); age (range/mean)) | Methodology | Perspective | Intervention | Objective of the study | Details of the technology |
|---|---|---|---|---|---|---|---|
| S. M. Andrew. s et al., 2017 [37] | USA | n = 18; 100; 60 | Qualitative; semi-structured interviews | Patients and HCPs (Home Telehealth Nurses) | Asynchronous (app) | To refine the intervention and inform the delivery of the intervention in other settings, by examining the participants' experiences. | This was a six-month telemedicine intervention in which HT nurses delivered the intervention's contents to the participants via biweekly calls. During each call, the nurses and the participants reviewed blood glucose, medications, and medication adherence. Nurses delivered self-management support on topics such as managing hypoglycemia. Following each encounter, a study physician reviewed patients' blood glucose data and recommended medication changes as indicated, and HT nurses implemented these recommendations. |
| J. Hall et al., 2018 [39] | UK (Scotland) | n = 15; NA; 18+ | Mixed methods, prospective; focus groups | Patients | Asynchronous (website) | To explore perceptions of diabetes knowledge, diabetes education, and uses of technology. | NA |
| L. Poppe et al., 2018 [40] | Belgium | n = 21; 61.9; 57–81 | Qualitative; semi-structured interviews | Patients | Asynchronous (website) | To assess participants' opinions regarding the usefulness of the implemented self-regulation techniques, the design of the programme, as well as the participants' knowledge regarding physical activity and sedentary behavior. | 'MyPlan 2.0' is a self-regulation-based eHealth intervention that targets physical activity and sedentary behavior. The website offers five sessions during which users can learn more about the beneficial effects of being less sedentary or more physically active via tips and quizzes, get feedback on their current levels of physical activity or sedentary behavior using a questionnaire, set their own goals for the coming week, search solutions for potential barriers, think about possible ways to keep track of their behavior changes, read about tips and tricks to become more physically active or less sedentary, and evaluate their behavior change processes each week. After an interval of one week, each user receives an email reminding him/her to start the following session. |

(Continued)

**Table 1.** (Continued)

| Author, year | Country | Participants (number; gender (male %); age (range/mean)) | Methodology | Perspective | Intervention | Objective of the study | Details of the technology |
|---|---|---|---|---|---|---|---|
| K. M. Smith et al., 2018 [41] | USA | NA; NA; NA | Qualitative; semi-structured interviews, focus groups | HCPs (nurses, nurse managers, and PCTs) | Asynchronous (web-based*) | To examine barriers and facilitators of integrating web-based, iPad- delivered diabetes survival skills education (DSSE) into the nursing inpatient unit workflow. | The Diabetes to Go program provides an adaptive learning approach and has been effective in improving medication adherence. The program was delivered in English on a web-enabled device. The patients first logged into the delivery platform and completed a 15-item validated survey to assess their knowledge of diabetes survival skills. Responses to the individual survey questions were then used to direct patients to video contents based on their personal knowledge deficits. |
| Pal K et al., 2018 [43] | UK (England) | n = 20; 60; 56.8 | Qualitative; focus groups | Patients | Asynchronous (website) | To explore patients' perspectives on unmet needs for self-management and support and the role of DHI in adults living with type 2 diabetes. | NA |
| L Desveaux et al., 2018 [45] | Canada | n = 13; 61.5; 32–67 | Qualitative; semi-structured interviews | Patients | Asynchronous (app) | To evaluate a web-based solution for improving self-management in type 2 diabetes to identify key combinations of contextual variables and mechanisms of action that explain for whom the solution worked best and in what circumstances. | The intervention is a commercially available app designed to serve as a web-based coach for patients with T2DM. This allowed participants to enter a range of baseline clinical information in addition to ongoing data related to diabetes management, including blood glucose values, daily medications, food intakes, and activity levels. The app analyzes inputted data to provide tailored messages to coach the participants with respect to their diabetes management. The participants also had the option of emailing reports to members of their care teams via the app, which provided them with an overview of the inputted data over periods specified by the participants. |
| L Kelly et al.; 2018 [44] | UK (England) | n = 15; 33.3; 55.4 | Qualitative; semi-structured interviews | Patients | Asynchronous (web-based) | To understand the impact of using web-based and mobile technologies to support the management of type 2 diabetes. | NA |

NA = not available; HCPs = healthcare professionals; PCTs = patient care technicians; DVD = digital versatile disc

*unspecified in the article.

The quality of the included articles by CASP can be found in S2 Appendix. In the thematic synthesis, two analytical themes were generated: i) *features and aspects of the intervention* and ii) *patient's experiences and perceptions*. From the analytical themes, seven descriptive themes were derived. There were four themes under features and aspects of the interventions including: *i) accessibility of the interventions, ii) mixed views on the technology, iii) applying self-management interventions*, and *iv) observations of HCPs*. Under patients' experiences and perceptions, there were three themes, including: *i) patient's motivation to use the interventions, ii) patients' personal attributes*, and *iii) views on the support received*.

## Features and aspects of the interventions

**Accessibility of the interventions.**   Patients preferred interventions with easy navigations, allowing them to know where to find the information that they were looking for, thus increasing their ease of use [33, 42]. They also appreciated the information being concise [40]. A lack of medical jargon facilitated information transfer for patients and enabled those with poorer grasp of language to benefit from the intervention [34]. However, some HCPs were concerned about the mismatch between the complexity of the content and the target audience's health literacy levels, where only those with better health literacy would benefit from the intervention [41].

Across the studies, patients reported technical difficulties while using the intervention. They reported that they had problems with devices [31], features of the intervention [31, 33, 37, 41], and editing their data [33, 37, 44]. They felt that the younger generation would be more familiar with online communications [33], which were common among technology-based interventions.

*"Um, but cause I did go in and I did try and do the tracking and I think cause I thought that was on an ongoing basis was the most useful part of it. But it was kind of a pain in the neck to use it. . . and kind of a pain in the ass getting where I wanted to go. I put some information and I wanted to delete it and I don't know if I ever succeeded in getting rid of it".* [33]

Patients with limited web access due to poor infrastructure or due to personal reasons were unable to access the interventions [33, 36, 37]. Some had issues with the cost of the interventions, believing that these should be free of charge [42]. This was especially the case for web-based interventions that required smartphones or computers to be accessed [42].

**Mixed views on the technology.**   There were mixed views regarding the information available through the intervention. Patients appreciated the fact that there was new information available that they did not know before [35, 45]. It helped some to gain a better understanding of their symptoms and they welcomed the fact that the information was available to them whenever they needed it [35–38, 44]. This instant availability of information promoted uses of the interventions to the patients [32, 35, 38]. Patients with prior knowledge of self-care practices did not find the interventions useful and hence they did not use them [40]. The perception of excessive information was a deterrent to some patients, reducing the interventions' usage [32, 43].

Patients had varying views regarding the trustworthiness of the information that they could find through the intervention. Patients especially liked the fact that they could get prompt, tailored advice from qualified persons [32, 42]. They used the interventions as adjuncts to their self-management education and found these to be "authoritative sources" since these were compiled, updated and monitored by HCPs [33, 39, 42]. This increased their confidence in the interventions and encouraged them to use them. In contrast, when lay persons were the ones

giving advice on community platforms without being monitored by HCPs, patients found the information unreliable [33].

> "I find for the most part it's the blind leading the blind. I guess this one is being moderated but by and large you have a bunch of people who don't know anything kind of spewing forth" - 54-year old woman [33].

Patients appreciated when the interventions were tailored to fit their specific circumstances [36, 42], and desired for the information to be more relevant to them when it was not [36]. Customized education has more applicability to a patient's life and will increase engagement with an intervention [42]. However, some felt that the information available was not relevant to real-life applications [33]. Some interventions were not tailored to the target groups, making these difficult to use and deterring them from using these [32, 38, 45].

Both patients and HCPs noted that there was variability in the information available from different sources. There was no standardization of the content and timing of education delivered to patients [41]. This lack of standardization was also evident to patients since they found different information on different websites [36] and were hence deterred from using the interventions.

Notifications for use effectively reminded patients of their diabetes and to use the interventions to manage their conditions [42, 44], acting as a facilitators for the intervention. Patients liked reminders when there was new information present through the interventions [36], since it made them more mindful about and act on their conditions [40].

**Applying self-management interventions.** Patients' views of the information were important factors in promoting the use of the interventions. Patients stated that they used the information as a motivator for change [44]. Information acted as a trigger for behavioral changes as patients learned new facts about their conditions and how to manage them [32, 35, 44, 45]. They recognized the value of self-care measures for diabetes through the intervention and were inspired to apply the information that they learned [38]. As a result of the information, they took their conditions more seriously [35].

> *"It's broadened my mind about everything. So, it's opened things up to me that I wouldn't have. . . if I'd have just gone on in my own little way, I would still be doing the same things so it has changed me, definitely, and I hope for the better."* [35]

Patients used the interventions to address specific concerns and found detailed information regarding these concerns [33]. The use of the information to inspire change and address specific needs facilitated the use of an intervention as a source of information for diabetes self-education.

Patients were supportive of technologies that allowed them to receive support from members of their healthcare teams or diabetes program staff [32, 42]. They expressed strong desires to share their progress with their HCPs either electronically or via a computer printouts [42–44]. Features of an application that facilitated communication with their HCPs were deemed attractive to patients [33].

**Observations of HCPs.** Various HCPs, including nurses, nurse managers and patient care technicians were in favor of using technologies for DSME as well [37]. Nurses strongly felt that there could be better integrations of education into workflows [41, 43]. They also supported DSME involving technology since it helped to provide information and convince patients of the benefits of treatments [34]. However, a lack of integration with the workflows on separate devices from the hospital systems made implementations by the HCPs difficult due to a lack of time to conduct and document them [41]. Other stakeholders, such as hospital

workers, had logistical concerns of cost, infection control, and safekeeping of the devices used for education within the hospitals [41].

## Patients' experiences and perceptions

**Patients' motivations to use the interventions.** Many patients wanted more information regarding the control of their illnesses [35, 36, 43, 45]. They independently sourced for more knowledge about diabetes to gain a sense of 'control' over their diseases and the impacts on their lives [32, 37, 43]. This group found the interventions useful in increasing their awareness of diabetes and its management [31, 37, 45].

Patients described that the interventions acted as a support and increased their responsibilities towards disease management [38, 45]. They also realized what their goals of diabetes management were and became more motivated to control their diseases as results of the interventions [37, 38, 40], and this motivation resulted in increased uses of the intervention [45]. Conversely, a lack of motivation towards the management of their diseases was the key hindrance to patients from seeking information [33, 37, 43, 45]. Some felt that it was pointless to manage the disease and that the complications were going to manifest regardless of any action taken [32, 33, 43] and hence did not attempt using DSME interventions.

*"I just find that all of these complications are so predestined, that no matter what you do, you are going to get these."* [32]

**Patients' personal attributes.** Increased use of the internet had reinforcing effects on patients such that they became more comfortable with using technology and their uses of technology-based interventions increased [33]. However, there were also some patients who had poor competence with technology [33, 38], which prevented them from using the interventions. HCPs noted that few patients could independently navigate the technology-based DSME due to poor literacy, language barriers, physical disabilities, a lack of technical skills, differences in learning needs and a lack of interest [41]. These HCPs proposed that alternative formats for program delivery should be available to accommodate these patients [41].

Another hindering factor against the use of technology-based DSME interventions was that patients had difficulty in finding balance between managing their diabetes and other aspects of life [33, 43, 45]. The struggle to balance various aspects of life left little time for patients to spend on educating themselves about diabetes using the technology-based interventions [33, 36, 37].

*"I go back and click on that date and enter all my sugars and meds and what not [all at once]. It's a lot easier than doing it daily—doing it daily it just eats up so much of my time. I only get a half hour lunch break at work usually. . .I don't want to spend my time fussing with it."* [45]

**Views on the support received.** Online forums and chats allowed patients to share their experiences, exchange practical advice, and rely on one another for social and emotional support [32, 33, 35, 42, 43]. Having people with shared experiences of living with diabetes also allowed patients to gain acceptance with each other [31, 38] and fostered a sense of community [32, 33, 44], which was lacking to some in their everyday lives [35]. Patients also liked the anonymity that allowed them to ask peers or experts questions to their peers or experts freely without judgement [33]. The ability to learn from others with previous experiences attracted patients to the interventions that incorporated such interactive features [35, 39].

*"Just that you had. . . somebody that's been through it like when you come to the group you're talking to people you know and you're. . . picking up wee bits and pieces."* [39]

The sense of community made patients more willing to use the interventions to support their self-management education [42].

*"So, I think learning to develop your support systems is extremely important for a diabetic. And that having a forum where even if you don't have a lot of people in your life that you can talk to about this, but having a forum where maybe you can go on and have an online community can be very helpful.".* [33]

However, there were patients who were not willing to take part in online forums since they did not want to share and preferred to keep their privacy [32, 42]. Some were self-conscious about people's judgements if they asked foolish questions [33] and did not participate in the online discussions.

Some patients also felt that there was a lack of social support online. Some preferred real life contacts to make meaningful conversations, deeming them more engaging and interactive [33, 42]. Others felt that there were disconnects between their opinions and those of others on online communities, often leading to frustration [35]. They were unable to relate to what others had shared and hence felt detached and preferred not to use these interventions.

## Discussion

To the best of our knowledge this is the first qualitative systematic review that examined patients' and HCP's perceptions towards technology-assisted DSME. The findings fit into two broad themes: 1) features and aspects of the interventions and 2) patients' experiences and perceptions when interacting with the interventions. The results reflect that technology-assisted DSME has the potential to adhere to the four guiding principles from the American Diabetes Association Standards of Medical Care in Diabetes care algorithm [10], such as information sharing [33, 35–43], psychosocial and behavioral support [31–39, 42, 43], coordination of care [37, 38, 40, 41, 43, 44] and engagement [32, 42, 43]. As per the existing evidence, patients play key roles in self-care in managing their own chronic diseases [46, 47]. Therefore, motivating them through DSME can be the main focus of the HCPs. While the included technology-based interventions were hosted on different platforms, after an examination of the included articles, it was found that these shared similar features and that patients across the various platforms expressed similar opinions and concerns. Hence, their sentiments had been taken as a whole in this synthesis.

This synthesis highlights that accessibility to technology-assisted DSME is one of the major facilitators for its use, which concurs with previous studies that technology-assisted DSME has the potential to enable patients' self-care through anonymous deliveries of personalized contents at multiple locations and at convenient times [48]. The convenience of technology can overcome the multiple barriers to accessing DSME, such as distance, time, financial constraints and limited primary care resources, which concur with the findings from other studies [49–51]. Moreover, literature has shown that increased access to DSME and HCPs, whether in-person or electronic, can improve diabetes knowledge and self-efficacy [52]. Therefore, it bodes well for an evidence-based and current technology-assisted DSME to complement HCP visits in the management of patients [53].

In this review, it was found that patients had mixed views regarding technology-assisted DSME. On the positive side, patients liked that technology-based education provides anonymous, timely, useful, and up-to-date information, and generally preferred interactive platforms to exchange information with their HCPs and platforms that had technology-based prompts. These perspectives are supported by previous studies that showed that patients' willingness to

use the technology platform was influenced by intervention designs such as having peer and counsellor support, emails or phone contacts and website updates [54], the ability to pace their own learning [55] or choose the complexity modes of information delivery based on literacy levels [56], and technology-based prompts [57]. In addition, similar positive views from this synthesis were reinforced by users' perceptions that include accepting the technology's effectiveness, usefulness, and enjoyment, which were mediated by trust [54, 58]. Therefore, this review demonstrates that a flexible and trusted technology platform with appropriate support and interactions is a facilitator for patients to use a technology-assisted DSME. This review discovered that motivated patients who were technology savvy and those who received support from HCPs and peers were more likely to use technology-assisted DSME platforms [59]. Patients who were more accepting to use the technology were those who were more motivated to take charge of improving their conditions, who trusted the technology and familiarised themselves with it, findings which concurred with previous studies [60, 61].

On the other hand, the results of this review showed that barriers to using technology-assisted DSME include perceptions of time constraints [33, 36, 37, 43], costs involved [42], poor motivation [32, 33, 37, 43, 45], and emotional distress or depression [32, 33, 43], which are largely consistent with previous studies about resistance in adoptions of novel technologies in patient-centered practice [60]. Other barriers included resistance to didactic platforms that disregarded patient's prior knowledge or health literacy levels [40], a lack of standardization of the information provided from different sources [36, 41], healthcare settings with limited technology infrastructure [33, 36, 37, 42], and poor integration of technology into work processes for nurses [41, 43], which are similar to concerns that were raised in the existing literature [60, 62]. This review also found that technical difficulties and concerns about inadequacies of technical skills are common barriers that hinder the use of technology [31, 33, 37, 41, 44], a finding supported by the literature [63]. Therefore, providing initial financial and technical support for patients such as vouchers for purchasing devices or courses for basic digital skills after enrolment into technology-assisted diabetes education may facilitate its use. For HCPs, the barrier of the technologies not being incorporated into routine work flows was similarly supported by the literature which advocates for integrations of the interventions into clinical workloads to ensure the maximal effectiveness of the technologies [64].

This review serves to remind stakeholders that technologies should assist and not hinder care delivery nor replace more direct human contacts and communications when required. Patients and HCPs indicated their views and preferences for effective, feasible, and acceptable technology features with options for selecting stratified delivery modalities that range from more interactive platforms to more static ones that offer facts and information based on each patient's health literacy, technical readiness, and privacy needs [60, 62, 65, 66].

## Implications for future research and practice

This study provides the basis for future research to evaluate the acceptability of technology platforms using standardized quantitative measures in larger scale studies to better inform developers. While qualitative evidence examines the users' experiences and facilitates the understanding of the behavioral elements of an intervention [23], quantitative data assesses the effectiveness of the intervention. Hence, larger scale original mixed method studies that provide both qualitative and quantitative evidence across a diverse range of patients will be highly informative in providing users' perceptions, as well as in addressing the impact of technology on clinical practice, healthcare costs to patients and the society, and patients' physiological outcomes. Further studies can be conducted to include the views of other formal or informal caregivers who are involved in the care of patients with type 2 diabetes, such as family

members and community health workers. From the quality assessment conducted of current literature, it was found that there was insufficient information regarding the recruitment strategies and the relationships between the researcher and the participants. By paying careful attention to these factors, future research can prove to be more valid and reliable. Additionally, there are currently limited qualitative studies on the HCPs' views about technology-assisted DSME, which future research can focus on.

## Limitations

The limitations of this review are as follows. Firstly, only English language publications were considered in this review. Secondly, while an attempt was made to include opinions from a diverse range of patients, the majority of the included studies focused on Western countries with well-developed economies that could support technology implementation. Within and across these studies, there were limited sample sizes and few variations in patient characteristics, hindering our ability to use diversity as a variable in patients' perceptions. With a small sample size, it is difficult to ascertain that the views from these patients were those held by the general user base. Thirdly, a large proportion of the studies in this paper commented on pilot technologies that have not yet become the standard of care. Additionally, the majority of the included articles involved web-based technology or websites. Hence, views regarding other forms of technology may not be evenly represented. While there was a large range in the ages of the patients in the included articles, age acts as a confounder in the perception of technology [60], with older generations often being averse to technology [60]. Lastly, while it would have been ideal to include views of all stakeholders involved, we were unable to find opinions from patients' caregivers, family members, community health workers and their families, hospital administrators, funders and legislators, even though they play an integral role in the education and care of a person with diabetes [67, 68].

## Conclusion

The purpose of this review was to analyze empirical evidence to date on the perspectives of patients and HCPs on receiving and delivering technology-assisted DSME. Technology-assisted DSME efforts appear to possess both positive and negative aspects as perceived by patients and HCPs. This review demonstrates that a positive patient mindset about education and the technology, accompanied by accurate, interactive, and timely information exchanges and provisions as well as support from HCPs or peers are key advocating factors for technology-assisted DSME. Technology platforms should be user-friendly, intuitive to use, and cater to older persons who may not be so technology savvy. Technical training and providing support to patients and caregivers who are less technology-savvy will go a long way to ensure the continuing meaningful use of technology-assisted DSME platforms.

## Supporting information

**S1 Checklist. PRISMA 2009 checklist DSME.**
(DOC)

**S1 Appendix.**
(DOCX)

**S2 Appendix.**
(DOCX)

**S3 Appendix.**
(DOCX)

## Author Contributions

**Conceptualization:** Lay Hoon Goh.

**Data curation:** Sneha Rajiv Jain, Yuan Sui, Cheng Han Ng.

**Formal analysis:** Cheng Han Ng.

**Funding acquisition:** Lay Hoon Goh.

**Methodology:** Shefaly Shorey.

**Project administration:** Cheng Han Ng.

**Supervision:** Shefaly Shorey.

**Writing – original draft:** Sneha Rajiv Jain, Yuan Sui.

**Writing – review & editing:** Zhi Xiong Chen, Lay Hoon Goh, Shefaly Shorey.

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
