## [Decision Letter · Decision Letter 0]

31 Mar 2020

PONE-D-20-02347

Perspectives Towards Technology-assisted Diabetes Self-Management Education. A Qualitative Systematic Review

PLOS ONE

Dear Dr. Shorey,

Thank you for submitting your manuscript to PLOS ONE. After careful consideration, we feel that it has merit but does not fully meet PLOS ONE’s publication criteria as it currently stands. Therefore, we invite you to submit a revised version of the manuscript that addresses the points raised during the review process.

We would appreciate receiving your revised manuscript by May 15 2020 11:59PM. To enhance the reproducibility of your results, we recommend that if applicable you deposit your laboratory protocols in protocols.io, where a protocol can be assigned its own identifier (DOI) such that it can be cited independently in the future. For instructions see: http://journals.plos.org/plosone/s/submission-guidelines#loc-laboratory-protocols

We look forward to receiving your revised manuscript.

Kind regards,

Fiona Harris, PhD

Academic Editor

PLOS ONE

Additional Editor Comments (if provided):

All three reviewers note the importance of this topic and note that this is an interesting paper of interest to this journal. However there are major revisions required before this would be suitable for publication. The reviewers have close read this manuscript and you should consider their recommendations as compulsory revisions. In summary, the paper requires:

1. the rationale should be strengthened in line with comments from Reviewers 2 & 3.

2. methods require some clarification: regarding screening and selection processes; inclusion/exclusion criteria; coding, analysis and inter rater reliability.

3. Editorial revisions are required throughout the manuscript in line with the reviewers' comments. Please address all points made by reviewers, who have close read the manuscript. You also would be advised to secure the services of a proof reading/editorial service in order to meet the publication standards of this journal since the journal does not proof read manuscripts. In addition to the language corrections, you also need to address the lack of supporting citations highlighted by reviewer 2.

Journal Requirements:

2. In the methods, please describe how risk of bias was assessed in individual studies (including specification of whether this was done at the study or outcome level, or both, and the specific test employed, such as the I^2 statistic), and how this information was used in any data synthesis.

In addition, please specify any assessment of risk of bias that may affect the cumulative evidence (e.g., publication bias, selective reporting within studies). Please ensure that the specific method of assessment (funnel plot, Egger's test, Begg's test, etc) is mentioned.

Reviewers' comments:

Reviewer's Responses to Questions

**Comments to the Author**

1. Is the manuscript technically sound, and do the data support the conclusions?

Reviewer #1: Partly

Reviewer #2: Yes

Reviewer #3: Partly

2. Has the statistical analysis been performed appropriately and rigorously? 

Reviewer #1: N/A

Reviewer #2: Yes

Reviewer #3: N/A

3. Have the authors made all data underlying the findings in their manuscript fully available?

Reviewer #1: Yes

Reviewer #2: Yes

Reviewer #3: No

4. Is the manuscript presented in an intelligible fashion and written in standard English?

Reviewer #1: Yes

Reviewer #2: Yes

Reviewer #3: No

5. Review Comments to the Author

Reviewer #1: An interesting and important topic. There needs to be more demonstration that the review is Protocol driven with a clearer PICO. In particular there is no rationale for including type 2 Diabetes and not type 1. The search strategy should note that review databases were checked, for example DARE.

Implications for future research is misleading: it states that mixed methods studies would be highly informative. As this type of study has been included in your review it's not clear if quantitative review is being suggested.

There are some grammatical and formatting errors throughout and there needs to be more attention to the use of language, including the discussion in first person which does not read well.

Reviewer #2: Thank you for the opportunity to review the current study. The authors’ systematic review and qualitative synthesis explores the perspectives of key stakeholders on diabetes management, a topic which, I believe, falls within the scope of PLOS ONE and is both timely and interesting. The introduction lays out a rationale for the study and the methods are clear and reproducible. The results appear measured and are largely well put into context in the discussion. I hope that the below comments are constructive and may serve to strengthen the paper.

Major Comments

1. The introduction provides good context around DSME, the move to tech-based interventions and the lack of synthesis of the qualitative research. However, I would have liked to have seen more verifiable support for the arguments being made. For example, the introductory paragraph (Lines 51-57) contains no citations to support the concept that diabetes management is an “art” or to highlight the importance of psycho-socio-economic-cultural-behavioural factors. Overall, many claims made throughout the introduction could be linked to citations or made clearer. For example in Lines 90-92, “Previous literature has shown that increased use of technological interventions was associated with greater improvements in outcomes.” It would be useful to note which types of interventions (web sites, text messaging, mobile health, etc) have been studied and which outcomes (HbA1c, weight management, lifestyle changes, etc) have been improved (and by how much).

2. In Line 92 of the introduction, there is a transition made in the authors’ argument. The point is made that technology-based interventions are improving outcomes. The authors then report that no effort has been made to summarise the qualitative literature on this topic. I think the connection between these two ideas could be strengthened. By this I mean: what is the explicit rationale for studying and synthesising the perspectives of stakeholders on technology-based interventions? If existing interventions are inadequate, additional stakeholder input might be useful for improving these interventions or it may be useful to argue for user input to ensure best practice guidelines are being followed. For example, justification for stakeholder involvement might be found in the work of Lucy Yardley and colleagues (a co-author on one of the included studies) who has argued for a person-based approach to digital health interventions. Alternatively, it might be useful to make the case that “identifying existing evidence” is key to intervention development according to the UK’s Medical Research Council. See citations below:

• Yardley, L., Morrison, L., Bradbury, K. and Muller, I., 2015. The person-based approach to intervention development: application to digital health-related behavior change interventions. Journal of medical Internet research, 17(1), p.e30.

• Craig, P., Dieppe, P., Macintyre, S., Michie, S., Nazareth, I. and Petticrew, M., 2008. Developing and evaluating complex interventions: the new Medical Research Council guidance. Bmj, 337, p.a1655.

3. In the methods section, PRISMA is referred to and is recommended by PLOS ONE. However, it may have been useful to have completed the Enhancing transparency in reporting the synthesis of qualitative research (ENTREQ) statement which might be more appropriate for a qualitative synthesis.

• Tong, A., Flemming, K., McInnes, E., Oliver, S. and Craig, J., 2012. Enhancing transparency in reporting the synthesis of qualitative research: ENTREQ. BMC medical research methodology, 12(1), p.181.

4. In the methods section (Lines 136-138), the quality appraisal of the included studies was briefly discussed. However, it does not appear that the rationale for this is mentioned. It would be useful for the reader to understand the authors’ rationale for including the appraisal, even if it is not to exclude studies. It might also be useful to comment on the appraisal in the discussion or where you make recommendations for future qualitative research around diabetes self-management. Across the included studies, many seem to fare poorly on two questions in particular:

• Has the relationship between researcher and participants been adequately considered?

• Was the recruitment strategy appropriate to the aims of the research?

5. In the results, at times, it can be difficult to identify which study is making a contribution. For me, it would be useful if each claim had a supporting citation. For example, which study supports the claims made in Lines 174-176?

6. As I was reading further about DSME in the Journal of Medical Internet Research and its sister journal JMIR Diabetes, I noted a couple of studies that might meet the inclusion criteria. It is entirely possible they were excluded, but I just wanted to highlight them. If they do not meet the criteria, it might be useful to make the inclusion criteria for DSME more explicit in the search strategy and screening section for readers (though it is mentioned briefly in the introduction).

• Kelly, L., Jenkinson, C. and Morley, D., 2018. Experiences of using web-based and mobile technologies to support self-management of type 2 diabetes: Qualitative study. JMIR diabetes, 3(2), p.e9.

• Desveaux, L., Shaw, J., Saragosa, M., Soobiah, C., Marani, H., Hensel, J., Agarwal, P., Onabajo, N., Bhatia, R.S. and Jeffs, L., 2018. A mobile app to improve self-management of individuals with type 2 diabetes: qualitative realist evaluation. Journal of medical Internet research, 20(3), p.e81.

7. In the discussion, it might be worth commenting on the impact of age in the main body of the discussion or in the limitations section. There seems to be diversity in the age across the studies. Yu et al 2014a, for example, ranges rom 20 to 79. Arguably, different age groups might perceive technology-based interventions differently.

8. Overall, the writing is appropriate and the authors’ message is understandable. However, it may be useful to consult a writing coach to make adjustments to the way sentences are written. For example, sometimes prepositions are left out or the tense of the sentence is written in the present where it would be better suited to the past. Having the manuscript reviewed may further improve its readability.

Minor Comments

9. Was any software used to support the coding process?

10. Table 1 is well laid out and enhances the readers’ understanding of the included studies. Two minor issues – it would be useful to have the citations numbered so the reader can tell more easily which study is making contributions in the results. Also, there is some inconsistency in the application of country names with regards the UK. Pal et al 2018 could likely be labelled “UK England” to match Patel et al 2015 and Hofmann et al 2018. Hall et al 2018 could have the “UK” attached to Scotland.

11. In Line 460-462, the citation is missing the journal title “BMC medical informatics and decision making”.

Reviewer #3: Dear Authors

Your review addresses an important issue, but I think it needs major revisions to meet the rigour required for publication. Kindly find the major revisions I recommend and other detailed comments attached.

6. PLOS authors have the option to publish the peer review history of their article (what does this mean?). If published, this will include your full peer review and any attached files.

Reviewer #1: No

Reviewer #2: No

Reviewer #3: Yes: Willem Odendaal

---

## [Author Response · Author response to Decision Letter 0]

7 May 2020

Response to Reviewers

Title: Patients’ and Healthcare Professionals’ Perspectives Towards Technology-assisted Diabetes Self-Management Education. A Qualitative Systematic Review

Manuscript ID: PONE-D-20-02347

Comments from Reviewer #1 

Reviewer’s Comments Response to Reviewers Lines

There needs to be more demonstration that the review is Protocol driven with a clearer PICO. In particular there is no rationale for including type 2 Diabetes and not type 1. Dear Reviewer, thank you for the comment. We have included more information about why we are focusing on technology-assisted DSME interventions for Type 2 diabetic adults in the introduction. Lines 59 – 69

The search strategy should note that review databases were checked, for example DARE. Dear Reviewer, thank you for the comment. Four review databases, namely, PROSPERO, Cochrane, Epistemonikos and McMasters Health evidence, were checked for this topic . No previous reviews were found regarding the views on the use of technology-assisted DSME. This is expanded on in the discussion section. Lines 128 – 130 

Implications for future research is misleading: it states that mixed methods studies would be highly informative. As this type of study has been included in your review it's not clear if quantitative review is being suggested. Dear Reviewer, thank you for the comment. We have clarified our suggestions for a future study involving a larger sized and a more diverse patient group, which provides qualitative views on the usage from the patients and HCPs and the quantitative effects of such interventions on the patients’ health. Lines 444 – 459 

There are some grammatical and formatting errors throughout and there needs to be more attention to the use of language, including the discussion in first person which does not read well. Dear reviewer, we apologise for the inconsistency and have made amendments throughout the manuscript. Throughout text

Comments from Reviewer #2 

Reviewer’s Comments Response to Reviewers Lines

The introduction provides good context around DSME, the move to tech-based interventions and the lack of synthesis of the qualitative research. However, I would have liked to have seen more verifiable support for the arguments being made. For example, the introductory paragraph (Lines 51-57) contains no citations to support the concept that diabetes management is an “art” or to highlight the importance of psycho-socio-economic-cultural-behavioural factors. Overall, many claims made throughout the introduction could be linked to citations or made clearer. For example in Lines 90-92, “Previous literature has shown that increased use of technological interventions was associated with greater improvements in outcomes.” It would be useful to note which types of interventions (web sites, text messaging, mobile health, etc) have been studied and which outcomes (HbA1c, weight management, lifestyle changes, etc) have been improved (and by how much). Dear Reviewer, thank you for the comment. We have added the relevant citations and supporting information for the arguments made in the introduction. Line 54; Line 84; Line 95; Lines 99 – 102 

In Line 92 of the introduction, there is a transition made in the authors’ argument. The point is made that technology-based interventions are improving outcomes. The authors then report that no effort has been made to summarise the qualitative literature on this topic. I think the connection between these two ideas could be strengthened. By this I mean: what is the explicit rationale for studying and synthesising the perspectives of stakeholders on technology-based interventions? If existing interventions are inadequate, additional stakeholder input might be useful for improving these interventions or it may be useful to argue for user input to ensure best practice guidelines are being followed. For example, justification for stakeholder involvement might be found in the work of Lucy Yardley and colleagues (a co-author on one of the included studies) who has argued for a person-based approach to digital health interventions. Alternatively, it might be useful to make the case that “identifying existing evidence” is key to intervention development according to the UK’s Medical Research Council. See citations below:

• Yardley, L., Morrison, L., Bradbury, K. and Muller, I., 2015. The person-based approach to intervention development: application to digital health-related behavior change interventions. Journal of medical Internet research, 17(1), p.e30.

• Craig, P., Dieppe, P., Macintyre, S., Michie, S., Nazareth, I. and Petticrew, M., 2008. Developing and evaluating complex interventions: the new Medical Research Council guidance. Bmj, 337, p.a1655. Dear Reviewer, thank you for the comment. We believe that the expansion of the rationale of examining qualitative literature would add value to the article and hence have amended the introduction accordingly. After examining the citations you have provided us, we believe that they help to build on the ideas presented in this paper and have thus included them as well. Lines 108 – 114

In the methods section, PRISMA is referred to and is recommended by PLOS ONE. However, it may have been useful to have completed the Enhancing transparency in reporting the synthesis of qualitative research (ENTREQ) statement which might be more appropriate for a qualitative synthesis.

• Tong, A., Flemming, K., McInnes, E., Oliver, S. and Craig, J., 2012. Enhancing transparency in reporting the synthesis of qualitative research: ENTREQ. BMC medical research methodology, 12(1), p.181. Dear Reviewer, thank you for the input. After considering the merits of ENTREQ, we have decided to use this statement as well and we have amended the methodology and results accordingly. Materials and Methods: Lines 121 – 122, 131, 143 – 149, 156, 161, 172 – 173

Results: 

Lines 217 – 220, 246 – 248, 276 – 278, 318 – 319, 337 – 339, 351 – 352, 357 – 360

In the methods section (Lines 136-138), the quality appraisal of the included studies was briefly discussed. However, it does not appear that the rationale for this is mentioned. It would be useful for the reader to understand the authors’ rationale for including the appraisal, even if it is not to exclude studies. It might also be useful to comment on the appraisal in the discussion or where you make recommendations for future qualitative research around diabetes self-management. Across the included studies, many seem to fare poorly on two questions in particular:

• Has the relationship between researcher and participants been adequately considered?

• Was the recruitment strategy appropriate to the aims of the research? Dear Reviewer, thank you for the comment. We conducted quality assessment using Critical Appraisal Skills Programme (CASP) to assess the validity of the recruitment of subjects, data collection, data analysis and results. Doing so improves the rigour of the synthesis and allows the assessment of the strength of the included articles. We have included the rationale in the methodology and expanded upon the recommendations for future research. 

 Lines 165 – 172 

In the results, at times, it can be difficult to identify which study is making a contribution. For me, it would be useful if each claim had a supporting citation. For example, which study supports the claims made in Lines 174-176? Dear Reviewer, thank you for the comment. We apologise for the oversight and have rectified the paper accordingly. Lines 207 – 209

As I was reading further about DSME in the Journal of Medical Internet Research and its sister journal JMIR Diabetes, I noted a couple of studies that might meet the inclusion criteria. It is entirely possible they were excluded, but I just wanted to highlight them. If they do not meet the criteria, it might be useful to make the inclusion criteria for DSME more explicit in the search strategy and screening section for readers (though it is mentioned briefly in the introduction).

• Kelly, L., Jenkinson, C. and Morley, D., 2018. Experiences of using web-based and mobile technologies to support self-management of type 2 diabetes: Qualitative study. JMIR diabetes, 3(2), p.e9.

• Desveaux, L., Shaw, J., Saragosa, M., Soobiah, C., Marani, H., Hensel, J., Agarwal, P., Onabajo, N., Bhatia, R.S. and Jeffs, L., 2018. A mobile app to improve self-management of individuals with type 2 diabetes: qualitative realist evaluation. Journal of medical Internet research, 20(3), p.e81. Dear Reviewer, thank you for the comment. We believe that the two suggested papers meet our inclusion criteria and hence have included them in our synthesis. Throughout text; Lines 174 – 434 

In the discussion, it might be worth commenting on the impact of age in the main body of the discussion or in the limitations section. There seems to be diversity in the age across the studies. Yu et al 2014a, for example, ranges from 20 to 79. Arguably, different age groups might perceive technology-based interventions differently. Dear Reviewer, thank you for the comment. We have taken this into account and expanded on this in the limitations section of the discussion. Lines 472 – 474 

Overall, the writing is appropriate and the authors’ message is understandable. However, it may be useful to consult a writing coach to make adjustments to the way sentences are written. For example, sometimes prepositions are left out or the tense of the sentence is written in the present where it would be better suited to the past. Having the manuscript reviewed may further improve its readability. Dear reviewer, we apologise for the inconsistency and have made amendments throughout the manuscript. Throughout text

Was any software used to support the coding process? Dear Reviewer, thank you for the comment. The coding process was done independently by the authors on an Excel sheet and a preformed data sheet. Lined 149

Table 1 is well laid out and enhances the readers’ understanding of the included studies. Two minor issues – it would be useful to have the citations numbered so the reader can tell more easily which study is making contributions in the results. Also, there is some inconsistency in the application of country names with regards the UK. Pal et al 2018 could likely be labelled “UK England” to match Patel et al 2015 and Hofmann et al 2018. Hall et al 2018 could have the “UK” attached to Scotland. Dear Reviewer, thank you for the comment. We apologise for the oversight and have rectified the paper accordingly. Table 1

In Line 460-462, the citation is missing the journal title “BMC medical informatics and decision making”. Dear Reviewer, thank you for the comment. We apologise for the oversight and have rectified the paper accordingly. Lines 608 – 611

Comments from Reviewer #3

Reviewer’s Comments Response to Reviewer Line

Major Revisions 

Provide a rationale for why the different technologies are lumped together. Is there evidence that technology type is not a variable to consider in understanding user perceptions and experiences? 

• Is an App on a phone (ref 22) similar enough to a website (in nine of the 13 included studies), and a website similar enough to telehealth (ref 23) to assume that blanket conclusions can be drawn across the types of technology? 

• Linked to this is a blanket statement (line 180) that “Some had issues with the cost of the intervention”. If it applied to all the technologies, say so, or tell the reader which technologies raised these concerns. Dear Reviewer, thank you for your comment. While we agree that differences exist between technology types, after examination of our included articles, we found that there can be many similarities drawn across the different platforms. Hence, we were able to group these sentiments together in our synthesis. This has been expanded upon in our discussion. Lines 224 – 225; 382 – 386; 

It appears that findings from Biernatzki et al (Information needs in people with diabetes mellitus: a systematic review. Syst Rev 7, 27: https://doi.org/10.1186/s13643-018-0690-0) suggest that there may be differences between diverse diabetes populations regarding information needs which in turn may inform patients’ technology preferences. Please provide a rational for not using diverse patient groups as a variable in perceptions and experiences. Dear Reviewer, thank you for the comment. Out of the papers that met our inclusion criteria, it was found that these studies largely examined views of those from Western populations, and across the studies included, the patient characteristics are homogeneous. Hence, we were unable to use diverse patient groups as a variable in perceptions and experiences. This is expanded upon in the limitations section of the discussion. Lines 446 – 450; 463 – 467

Points 1 and 2 should inform a more nuanced Discussion and Conclusions. Dear Reviewer, thank you for the comment. We believe that these points hold merit and have incorporated them into the discussion. Lines 382 – 474

Please link this review with effectiveness reviews. This should be addressed in the Background and Discussion, with a deliberate reference on how they complement each other. It will be interesting to know if their qualitative results can help understand effectiveness outcomes. Are the quantitative results of the effectiveness reviews, Reference 32 and 39, not of any relevance to understand the importance of patients’ perceptions and experiences? Dear Reviewer, thank you for the comment. We believe that this is a valid suggestion and have expanded on the possibility of studies linking the perceptions to the effectiveness in the ‘Implications for future research and practice’. 

Regarding quantitative results of mixed method reviews in this study, we believe that the sole focus of this review is to capture the barriers and facilitators to the use of technology-assisted DSME and hence have not included them in this review. However, future studies can examine the outcomes of these interventions to determine its effectiveness. Therefore, we have commented on a possibility of a future study in the ‘Implications for future research and practice’ section of the discussion. Lines 446 – 459 

Provide a rationale why they excluded the perceptions and experiences of informal caregivers such as family members of lay medical workers. I’m not expecting them to be included, but my view is that they play a big part in the self-management of a chronic condition, as healthcare workers, and hence their voices are important? Dear Reviewer, thank you for the comment. From the articles that met our inclusion criteria, we were unable to find the views of these informal caregivers and hence these groups were not included. However, these groups of people are vital in the care of those with chronic diseases like diabetes. Hence we believe that future studies should include their perspectives as well. We have expanded upon this in the ‘implications for future research and practice’ and ‘limitations’ section of our discussion. Lines 474 – 478

Provide more information regarding their methodology: 

• Provide the screening tool, i.e. the inclusion / exclusion criteria, in particular the participants and technology. 

• Did you have criteria regarding publication date and setting? 

• Detail who did the screening: was it the same two authors for title/abstract and full texts respectively, and if Yes, why? 

• Add more detail about the coding process: (i) who did it, and (ii) did the referred two authors coded each paper independently and then compared, or did they sat together, or did they do one or two together to draft a coding list, or did they use a different method. 

• Substantiate their claim that theirs is the 1st qualitative systematic review by providing evidence that they have searched systematic review data bases, such as: Cochrane library: https://www.cochranelibrary.com/ Epistemonikos: https://www.epistemonikos.org/ McMasters Health evidence: https://www.healthevidence.org/ Dear Reviewer, thank you for the comment. Based off of your recommendations, we have added more information about the inclusion/exclusion criteria, details of authors who did the screening (in line with the ENTREQ statement recommended), and the coding process in the Methodology. 

The claim that this is the first qualitative systematic review was checked in four systematic review databases, namely, PROSPERO, Cochrane, Epistemonikos and McMasters Health evidence. No previous reviews were found regarding the views on the use of technology-assisted DSME. We have added this statement in the Materials and Methods. Lines 126 – 173 

Please add quotes strategically that will clarify some of the results. 

 Dear Reviewer, thank you for the comment. We have included quotes from the included articles to better reflect the meaning of the results. Lines 217 – 220, 246 – 248, 276 – 278, 318 – 319, 337 – 339, 351 – 352, 357 – 360

Evidence that the manuscript was proof-read for language. There are too many instances, such as the following, that need a revision to clarify its meaning. 

• Lines 33-34: “Stakeholders had mixed views towards features of the technology-assisted interventions and patients’ personal qualities and providers’ concerns that affected their use of the interventions.” 

• Lines 309-10: “Since patients were unable to fulfill their preferred method of social support online and felt that these interventions were ineffective in creating peer support.” Dear Reviewer, thank you for the comment. We apologise for the oversight and have amended the errors. Lines 309 – 310 have been deleted from the manuscript. We also have edited the manuscript to fix the language. Line 34; throughout text

Important in Text Revisions 

Title: Add wording to the title that will make it clear that the review is about patients’ and healthcare workers’ perspectives. Dear Reviewer, thank you for the comment. We have amended the title accordingly. Line 1

At times you refer to healthcare workers (HCWs) as “professionals” (line 39); “various healthcare providers”, (line 61) which could include all cadres; “Nurses, nurse managers and patient care technicians” (line 247); and “hospital workers” (line 252). 

• Decide on a terminology and describe it clearly and use it consistently. Dear Reviewer, thank you for the comment. We apologise for the oversight. We have rectified this by standardising the use of the term ‘healthcare professional’ throughout the text. Throughout text 

The Discussion in lines 42-3 looks like a conclusion, and there is no conclusion in the abstract. Dear Reviewer, thank you for the recommendation. We have amended the abstract accordingly. Lines 43 – 46 

Add a space between the last word and in-text citation of a reference. Dear Reviewer, thank you for the comment. We have amended the paper accordingly. Throughout text 

Line 21: Add whose perspectives you are referring to. Dear Reviewer, thank you for the recommendation. We have amended the abstract accordingly. Line 21

Line 23: Add HCWs to the sentence, else it seems it is only about patients. Dear Reviewer, thank you for the recommendation. We have amended the abstract accordingly. Line 24

Line 33: Clarify who the stakeholders are, keeping in mind that they may include people other than patients and HCWs Dear Reviewer, thank you for the recommendation. We have amended the abstract accordingly. Line 34

Line 38: At first, I thought of “Community support” as in the communities where patients live, but in the manuscript, it is about fellow patients. Clarify this in line 38. Dear Reviewer, thank you for the comment. We have amended the abstract accordingly. Line 39 

Line 55: Add a reference. Dear Reviewer, thank you for the comment. We have added the appropriate citation for this claim. Line 54

Line 84: Clarify “caregivers”: is it only HCWs or does it include family members too? Dear Reviewer, thank you for the comment. We have rectified the section by clarifying which group this statement refers to. Line 93

Line 86: Add a reference. Dear Reviewer, thank you for the comment. We have added the appropriate citation for this claim. Line 95

Line 88: Clarify “disruptive”. Dear Reviewer, thank you for the comment. We have elaborated on the definition of the term in the text. Line 89 – 91 

Line 96: Clarify “stakeholders” Dear Reviewer, thank you for the comment. We have elaborated on which group the term ‘stakeholders’ is referring to in the text. Line 107

Line 111: What software did you use to screen, and how did you resolve the differences? Dear Reviewer, thank you for the comment. The citations were screened manually and two authors resolved the differences by consensus. We have added the appropriate information. Lines 131 – 134

Line 118: Did you include original, non-peer reviewed papers? Dear Reviewer, thank you for the comment. Only original, peer-reviewed papers were included. We have rectified the section as such. Line 141

Line 119: What is the difference between a ‘dissertation’ and ‘thesis’? Dear Reviewer, thank you for the comment. A dissertation and a thesis are made at different levels of academia. A thesis is written to show consolidation of knowledge at the master’s level while a dissertation is to contribute new knowledge at a doctoral level. NA

Line 120: Add the software you used. Dear Reviewer, thank you for the comment, Information regarding the software used has been added into the text. Line 144, 150

Line 121: “a senior author” could be a content expert outside the author team? Dear Reviewer, thank you for the comment. We have clarified which authors this sentence is referring to. Lines 145 – 146

Line 132: Which authors? Dear Reviewer, thank you for the comment. We have clarified which authors this sentence is referring to. Lines 161 

Line 138: Who was the “independent 3rd author?” ‘Independent’ from the appraisal but part of the team? Dear Reviewer, thank you for the comment. This author was also involved in this appraisal. We apologise for the oversight and have clarified this in the text. Line 173

Table 1: 

• Clarify how ‘web-based’ and ‘web-site’ differs in refer 18. 

• Provide a rationale why you choose age as a variable and not something like SES or education? Is there literature that links attitude towards technology with age? Dear Reviewer, thank you for the comment. Regarding the first clarification, the article did not provide more details about the intervention, apart from it being web-based. Hence, we are unable to elaborate on the difference between the two. 

In this table, age was not chosen as a variable, but rather to list out the baseline characteristics of patients involved in the studies. Table 1

Lines 174-76:

Add references and “with better health literacy will be able to …” should rather be in the discussion or rephrased as a result. Dear Reviewer, thank you for the comment. The sentence has been amended to better convey its intended meaning. Lines 207 – 209

Lines 181-82: Is this not a repeat of line 179-80? Dear Reviewer, thank you for the comment. We apologise for the oversight and have rectified the paper accordingly. The sentence has been deleted from the manuscript. -

Line 185: It can be understood as information about the intervention, for example how to use it. My read is that you rather mean the “information made available through the intervention.” If that the is the case, rephrase the sentence. Dear Reviewer, thank you for the comment. We have clarified the meaning of the sentence in line with your suggestion. Line 228

Lines 196-97: See my query regarding line 185. Dear Reviewer, thank you for the comment. We have clarified the meaning of the sentence in line with your suggestion. Line 238

Line 208: Add reference. Dear Reviewer, thank you for the comment. We have amended the text accordingly and have added the comment. Line 253

Lines 209-10: How is it different to lines 190-91? Dear Reviewer, thank you for the comment. We apologise for the oversight and have rectified the paper accordingly. Lines 209 – 210 have been deleted from the manuscript. - 

Line 213: What value is it adding as it repeats lines 216-17? Dear Reviewer, thank you for the comment. We apologise for the oversight and have rectified the paper accordingly. Lines 216 – 217 have been deleted from the manuscript. -

Line 215: “They felt…”: Assuming these are the older patients, do you have data on what the younger patients themselves said? Dear Reviewer, thank you for the comment. Unfortunately, from the included articles, we were unable to find opinions from younger users, hence have not included them in our findings. -

Line 227: See my query line 185. Dear Reviewer, thank you for the comment. We have clarified the meaning of the sentence in line with your suggestion. Line 265

Line 244: Add reference. Dear Reviewer, thank you for the comment. We have added the reference to the Line 289

Line 248: This theme is about health workers and not patients. Dear Reviewer, thank you for the comment. We apologise for the oversight and we have clarified which group of people this is referring to. Line 292

Line 249: Who are “they”: patients and nurses? Dear Reviewer, thank you for the comment. We apologise for the oversight and we have clarified which group of people this is referring to. Line 293

Line 249: Provide an example of “integration with the workflow”. Dear Reviewer, thank you for the suggestion. We have provided an example in the text, as per your recommendation. Line 296 – 298

Line 253: Were the technology used in the hospital? Dear Reviewer, thank you for the comment. We have amended the sentence to clarify its meaning. Line 299 – 300 

Line 260: A repeat of lines 188-89? Dear Reviewer, thank you for the comment. We apologise for the oversight and have rectified the paper accordingly. Line 260 has been deleted from the manuscript. - 

Lines 261-62: Reads like a conclusion and not result. Dear Reviewer, thank you for the comment. We apologise for the oversight and have rectified the paper accordingly. Lines 261 – 262 have been deleted from the manuscript. -

Lines 303-05: Is this the authors’ opinion or patients’? If the former, move it to the discussion. Dear Reviewer, thank you for the comment. We apologise for the oversight and have rectified the paper accordingly. Lines 303 – 305 have been removed from the manuscript. -

Lines 356-62: This is just a repeat of results, and should rather be discussed in relation to the literature on these issues. Dear Reviewer, thank you for the comments. Based off of your suggestion, we have presented our review’s findings in relation to existing information on these issues. Lines 418 – 441

Line 369: Please provide practical examples of how this support can be provided. Dear Reviewer, thank you for the comment. We have supplemented the suggestion of providing support with relevant examples. Lines 429 – 430 

Line 360: Clarify “providers”; see Query 1 above. Dear Reviewer, thank you for the comment. We have amended the sentence to clarify who ‘providers’ refers to. 

Lines 373-78: Please link this with existing literature. Dear Reviewer, thank you for the comment. The appropriate citations have been added to link this claim with existing literature. Line 441

Limitations: 

• Qualitative research is not about numbers, but all the included studies were small. I suggest it as a limitation in a more general sense of studies exploring the use of technology. 

• How many of the included studies reported on technology that have become standard care? My sense from the titles are that many of the technologies were being piloted. If that is indeed the case, I think it to be a limitation, because piloting is generally a world apart from standard practice, with the former often being better resourced with project staff who want to make the technology work. 

• Nine of the 13 included studies are about web-based technology or website technology. Is this not skewing the results?

• Clarify who the “administrators” are? Dear Reviewer, thank you for the input. We believe the your suggested limitations hold merit and have incorporated them into our limitations. Lines 462 – 478 

Line 398: Is there evidence in the results that HCWs were involved in delivering the DSME technology? Dear Reviewer, thank you for the comment. From our included articles, there was evidence that HCPs were involved in delivering the DSME through technology. This has been clarified in the results, under “Observations of HCPs”. Line 296 – 297 

Lines 400-03: Is it not that the “positive engagement” resulted from all the issues listed, and not preceded it? Dear Reviewer, thank you for the comment. We have rectified the sentence to better convey its intended meaning. Line 484 – 487

Line 409: This does not make sense as there are no abbreviations provided Dear Reviewer, thank you for the comment. We apologise for the oversight and have amended the section accordingly. Lines 493 – 495 

S1 Appendix: I’m missing “technology” in the search terms. If it is missing, is there a reason for not including it? Dear Reviewer, thank you for the comment. The exclusion of the term was based on the recommendation from a medical librarian. - 

Line 36: Use a synonym for “impetus” Dear Reviewer, thank you for the comment. We have edited the section accordingly. Line 36

Line 45: “was” rather than “is”, that is assuming that the study came to an end. Dear Reviewer, thank you for the comment. That statement has been removed in light of the fact that this paper is no longer funded by the grant. - 

Lines 69-72: Too long sentence. Dear Reviewer, thank you for the comment, we have amended the sentence accordingly, while still keeping with its meaning. Lines 77 – 79

Lines 126-131: Add an example of the coding > descriptive themes > analytic themes as an appendix. Dear Reviewer, thank you for the comment. We have added an appendix as per your suggestion. S3 Appendix

Line 143: Add the number of studies to the countries. Dear Reviewer, thank you for the comment. We have edited the results section accordingly. Lines 177 – 179

Table 1 

- Include the reference number below the author name 

- What about adding an appendix with detail of how the technology worked? think it will help the reader to better understand the results 

- Be consistent in ‘interviews’ in the Methodology column. Sometimes you used ‘interview’ 

- 1st time use of “HCP” requires writing in full. Dear Reviewer, thank you for the comment. We have edited Table 1 accordingly. We believe that your second suggestion holds merit and hence we have also included a column in Table 1 explaining the details of how the technology worked. We feel that it is better to include this information in Table 1 rather than the appendix since this information is vital in the reader’s understanding of how such interventions work. Table 1

Line 184: Rather just “Mixed views on the technology provided intervention” Dear Reviewer, thank you for the recommendation. We have edited the heading accordingly. Line 196, 227 

Line 201: “for diabetes self-management” seems redundant to me. Dear Reviewer, thank you for the recommendation. We have edited the section accordingly. Line 242

Lines 213-17: Does it not belong to “Accessibility of the Intervention”? Dear Reviewer, thank you for the comment. We believe that this paragraph belongs in the suggested section and we have edited the manuscript accordingly. Lines 211 – 215 

Line 230: Was there another application as to real life? Consider some like: ‘Applying self-management interventions. Dear Reviewer, thank you for the recommendation. We believe that this term better captures the meaning of this section and have amended the section accordingly. Lines 196 – 197, 268 

Line 254: Consider “perceptions” rather than “emotions”. Dear Reviewer, thank you for the recommendation. We believe that this term better captures the meaning of this section and have amended the section accordingly. Line 194, 302

Line 274: “had” rather than “has” Dear Reviewer, thank you for the comment. We apologise for the oversight and have rectified the paper accordingly. Line 322

Line 318: Will “perceptions” not work better than “emotions”? Dear Reviewer, thank you for the recommendation. We believe that this term better captures the meaning of this section and have amended the section accordingly. Line 376

Lines 322, 362: “previous literature” is not scholarly. Dear Reviewer, thank you for the comment. We apologise for the oversight, and have rectified the paper accordingly. Line 380, 426 

Line 365: “hiccups” not scholarly Dear Reviewer, thank you for the comment. We apologise for the oversight, and have rectified the paper accordingly. Line 427 

Lines 381-82: Consider adding “using standardised quantitative measures in larger scale studies, to better…”. Dear Reviewer, thank you for the suggestion. We believe that your suggestion adds value and hence have included it in the writing. Line 445

Lines 382-86: You are saying that mixed-methods are required, but 5/13 (38%) included studies used mixed methods. Why is this not enough? Dear Reviewer, thank you for the comment. The included mixed-method studies all have a small sample size and are hence unable to capture the general consensus regarding DSME interventions. We believe that larger scale mixed method studies would be able to understand the user experience and find the impacts of such interventions. We have thus clarified the sentence to bring out this meaning. Lines 446 – 452

Line 385: How is “biochemical” and “psychological” different? Dear Reviewer, thank you for the comment. We have amended the wording to better capture the intended meaning of the sentence. Line 452

Line 405: Which “group” are you referring to? It reads as if it is not diabetic patients but a group of these patients. Dear Reviewer, thank you for the comment. We have clarified which patients this line is referring to in the Conclusion. Line 488 – 489 

Fig 1: 36 studies were excluded because of quantitative analysis. What about the data collection? Consider using “quantitative methods”. Dear Reviewer, thank you for the comment. We apologise for the oversight and have rectified the paper accordingly. Fig 1

Appendix 2: Consider adding a column that details your overall assessment: ‘no/very minor concerns’ or ‘minor concerns’ or ‘moderate considerations’ or ‘serious concerns’ about the methodological rigour of a study. Dear Reviewer, thank you for the comment. We have added the suggested column into Appendix 2. Appendix 2

---

## [Decision Letter · Decision Letter 1]

13 Jul 2020

PONE-D-20-02347R1

Patients’ and Healthcare Professionals’ Perspectives Towards Technology-assisted Diabetes Self-Management Education. A Qualitative Systematic Review

PLOS ONE

Dear Dr. Shorey,

Thank you for submitting your manuscript to PLOS ONE. After careful consideration, we feel that it has merit but does not fully meet PLOS ONE’s publication criteria as it currently stands. Therefore, we invite you to submit a revised version of the manuscript that addresses the points raised during the review process.

We look forward to receiving your revised manuscript.

Kind regards,

Emily A Hurley, M.P.H., Ph.D.

Academic Editor

PLOS ONE

Reviewers' comments:

Reviewer's Responses to Questions

**Comments to the Author**

1. If the authors have adequately addressed your comments raised in a previous round of review and you feel that this manuscript is now acceptable for publication, you may indicate that here to bypass the “Comments to the Author” section, enter your conflict of interest statement in the “Confidential to Editor” section, and submit your "Accept" recommendation.

Reviewer #2: (No Response)

Reviewer #3: All comments have been addressed

2. Is the manuscript technically sound, and do the data support the conclusions?

Reviewer #2: Yes

Reviewer #3: Yes

3. Has the statistical analysis been performed appropriately and rigorously? 

Reviewer #2: N/A

Reviewer #3: N/A

4. Have the authors made all data underlying the findings in their manuscript fully available?

Reviewer #2: Yes

Reviewer #3: Yes

5. Is the manuscript presented in an intelligible fashion and written in standard English?

Reviewer #2: Yes

Reviewer #3: Yes

6. Review Comments to the Author

Reviewer #2: (No Response)

Reviewer #3: Dear Authors

Thanks for your comprehensive response to my comment.

A small matter:

Please explain to readers not familiar with technology jargon, what “disruptive technologies” mean. My understanding is that it refers to innovate technologies that create a "new market and value network and eventually disrupts an existing market and value network, displacing established market-leading firms, products, and alliances". If this is indeed what you meant with 'disruptive' please add, or if you have a different definition, please add that.

7. PLOS authors have the option to publish the peer review history of their article (what does this mean?). If published, this will include your full peer review and any attached files.

Reviewer #2: No

Reviewer #3: **Yes: **Willem Odendaal

---

## [Author Response · Author response to Decision Letter 1]

16 Jul 2020

Comments from Reviewer #3 

Reviewer’s Comments Response to Reviewers Lines

Thanks for your comprehensive response to my comment.

A small matter:

Please explain to readers not familiar with technology jargon, what “disruptive technologies” mean. My understanding is that it refers to innovate technologies that create a "new market and value network and eventually disrupts an existing market and value network, displacing established market-leading firms, products, and alliances". If this is indeed what you meant with 'disruptive' please add, or if you have a different definition, please add that. Dear Reviewer, thank you for the comment. We have added to the manuscript, detailing the meaning of disruptive technology in the context of diabetes self-management education. Lines 89-93

---

## [Editor Report · Decision Letter 2]

31 Jul 2020

Patients’ and Healthcare Professionals’ Perspectives Towards Technology-assisted Diabetes Self-Management Education. A Qualitative Systematic Review

PONE-D-20-02347R2

Dear Dr. Shorey,

We’re pleased to inform you that your manuscript has been judged scientifically suitable for publication and will be formally accepted for publication once it meets all outstanding technical requirements.

Kind regards,

Emily A Hurley, M.P.H., Ph.D.

Academic Editor

PLOS ONE
---

## [Editor Report · Acceptance letter]

6 Aug 2020

PONE-D-20-02347R2 

Patients’ and Healthcare Professionals’ Perspectives Towards Technology-assisted Diabetes Self-Management Education. A Qualitative Systematic Review 

Dear Dr. Shorey:

I'm pleased to inform you that your manuscript has been deemed suitable for publication in PLOS ONE. Congratulations! Your manuscript is now with our production department. 

Kind regards, 

on behalf of

Dr. Emily A Hurley 

Academic Editor

PLOS ONE